# MODULAR DEEP PROBABILISTIC PROGRAMMING

## ABSTRACT

Modularity is a key feature of deep learning libraries but has not been fully exploited for probabilistic programming. We propose to improve modularity of probabilistic programming language by offering not only plain probabilistic distributions but also sophisticated probabilistic model such as Bayesian non-parametric models as fundamental building blocks. We demonstrate this idea by presenting a modular probabilistic programming language MXFusion, which includes a new type of re-usable building blocks, called *probabilistic modules*. A probabilistic module consists of a set of random variables with associated probabilistic distributions and dedicated inference methods. Under the framework of variational inference, the pre-specified inference methods of individual probabilistic modules can be transparently used for inference of the whole probabilistic model. We demonstrate the power and convenience of probabilistic modules in MXFusion with various examples of Gaussian process models, which are evaluated with experiments on real data.

## 1  INTRODUCTION

Deep neural networks (DNNs) have seen great successful in various areas such as computer vision, speech recognition, and artificial intelligence. A major driver of such success is the rise of deep learning libraries, starting from Theano and following by many popular libraries such as TensorFlow, PyTorch. The unique benefits that these libraries bring are: (1) completely transparent training and prediction, (2) modular DNN construction with interchangeable components. A DNN component can vary dramatically in terms of size and complexity, ranging from a single matrix operator, to a highly complex deep neural network such as convolutional neural networks (CNN) or long short-term memory (LSTM). All the components share the same interface, which enables a user to replace one component by another with almost no changes on other parts of the code. In particular, modularity in the programming languages used by deep learning libraries revolutionizes the development, communication and deployment of DNNs. The ability to package a state-of-the-art implementation into a standard module and distribute enables rapid development with much lower chance of having bugs and performance issues comparing with re-implementing from scratch, which makes research results more reproducible. Communication about ongoing research method and results becomes much easier, as running a different experiment or modifying a network only requires to change a few lines of code. The standardized interface of components enables better testing and maintainability. It eases the transition of a research implementation into an industrial software.

Probabilistic programming language (PPL) is a similar domain specific programming language that aims at describing probabilistic models and automating inference in those models. Despite the different of DNNs and probabilistic models in mathematical formulations[1], a similar modularity structure exist in both probabilistic models and DNNs. However, the benefits of modularity has not been fully exploited in currently popular PPLs such as Stan (Carpenter et al., 2017), Edward (Tran et al., 2016), PyMC (Salvatier et al., 2016), which is mostly about re-using plain probabilistic distributions and (configurable) generic inference algorithms such as Markov Chain Monte Carlo (MCMC) or Stochastic Variational Inference (SVI).

In this paper, we propose to further exploit the modularity in probabilistic models by offering not only plain probabilistic distributions but also sophisticated probabilistic model such as Bayesian

---

[1] A DNN represents a deterministic function, while a probabilistic model describes a probabilistic distribution.

non-parametric models as fundamental building blocks in a PPL. This would bring the a lot of benefits of modularities that we enjoy with deep learning libraries and improve the usabilities of PPLs on real world problems. We demonstrate this idea by presenting a modular probabilistic programming language MXFusion, which includes a new type of re-usable building blocks for probabilistic models called *probabilistic modules*. A probabilistic module consists of a set of random variables with associated probabilistic distributions and dedicated inference methods designed specifically for efficiency and accuracy on that set of random variables. This allows to package a sophisticated probabilistic model with their proposed inference methods. The reason for packaging inference methods within a probabilistic module is due to the fact that the exact inference of most of probabilistic models are intractable. Given a specific probabilistic model, a generic approximate inference method that is applicable to a wide range of probabilistic models often performs worse than a dedicated inference method. Most of current PPLs only rely on generic inference methods such as MCMC or black-box variational inference. We propose a framework of probabilistic modules based on variational inference. With the proposed framework, one could seamlessly combine generic inference methods with dedicated inference methods, which bridges the performance gap.

The rest of the paper is organized as follows: Section 2 describes relevant ideas in the literature of PPL; Section 3 provides details of probabilistic modules; Section 4 and **??** presents probabilistic modules in MXFusion; Section 5 demonstrates experiments of presented models with real data.

## 2 RELATED WORKS

Probabilistic programming languages typically focus on offering an expressive programming language that describes probabilistic models and automating inference of probabilistic models through a "compilation" step. Due to the intractability of inference in probabilistic models, existing PPLs either focus on the expressiveness and employs a generic inference engine which suffers from scalability issues (Pfeffer, 2001; 2009; Goodman et al., 2012), or focus on efficiency by restricting down to a specific class of models and inference algorithms (Spiegelhalter et al., 1995; Murphy, 2001; Salvatier et al., 2015; Carpenter et al., 2016). By exploiting automatic-differentiation and hardware acceleration, Edward (Tran et al., 2017) and Pyro (Goodman, 2017) take a middle approach by offering the capability of customized inference methods, which is often referred to as an inference model. Both propose a clear separation between model and inference and typically rely on stochastic variational inference methods such as amortized inference.

The idea of encapsulating models and approximate inference programs in probabilistic modules has also been proposed by Cusumano-Towner & Mansinghka (2017), in which they demonstrate with Markov Chain Monte Carlo (MCMC) and Sequential Monte Carlo (SMC) inference programs. Venture and Anglican define inference as a collection of program fragments corresponding to local inference problems, although they do not support customizable posterior approximation (Mansinghka et al., 2014; Wood et al., 2014).

## 3 MODULARITY IN PROBABILISTIC PROGRAMMING

DNN has a very nice modularity property that is exploited by deep learning libraries, which is a result of function composition. The formulation of a function can be broken into the composition of several sub-functions, e.g., $f(x) = g_1(\cdots g_k(x))$. This enables a deep learning library to implement building blocks at various granularities as individual sub-functions. Then, a user can easily construct a DNN by putting these building blocks via function composition.

A similar modularity property also exists in probabilistic models, but in sightly different form. The joint probability distribution of a probabilistic model can often be decomposed into the product of the conditional distribution of individual variables with all the latent variables being marginalized out, e.g., $p(y|z) = \int_x p(y|x)p(x|z)$. This allows us to implement these individual conditional distributions as building blocks for a probabilistic model. So far, most of PPLs exploits this modularity and provides standard probabilistic distribution as building blocks. In this paper, we present an approach to provide more sophisticated probabilistic models as building blocks.

Inference for a DNN is mostly based on gradient optimization, in which a gradient can be computed transparently through automatic-differentiation. For a probabilistic model, inference is very chal-

lenging, as the exact inference for most of probabilistic models is intractable. Research has been focused on developing approximate inference or sampling methods for resolving this intractability. The literature of approximate inference methods can be broadly categorized into two groups: (1) proposing a better approximate inference method for a specific probabilistic model (2) developing a generic inference algorithm that can be applied to a family of probabilistic models as big as possible, sometimes being referred to as black-box inference. Both types of inference methods have received great attention and been developed by a lot of researchers. Obviously, given a specific model, a specialized inference method will always give a better performance than a generic inference method, as it is able to take advantages of the specific mathematical properties of the given model. Even among generic inference methods, a method only applicable to a small family of models performs better than a more generic method. For example, stochastic variational inference with reparameterization trick that is only applicable to a location-scale family of distributions produces a lower variance estimate of gradient than the score function approach that is applicable to mostly of distribution including discrete distributions.

Currently existing PPLs only use generic inference methods. Although those generic inference methods may be configurable, e.g., providing a customized variational posterior, it often performs worse for the probabilistic models, of which a specialized inference method exists. In MXFusion, we bridge the performance gap between generic and specialized inference methods by further exploiting the modularity of probabilistic models and offering *probabilistic modules* as building blocks. A *probabilistic module* encapsulates a sophisticated probabilistic model, e.g., a Gaussian process, and a set of specialized inference algorithms. With the latent variables of a probabilistic module being invisible from outside, a probabilistic module defines a conditional distribution over its exposed variables, which has no difference from a plain probabilistic distribution. A probabilistic module offers the same interface as a plain probabilistic distribution, so that an external inference method can transparently treat a probabilistic module as a plain probabilistic distribution. Internally, the interface of a probabilistic module is realized by calling a pre-specified inference method.

In the following section, we demonstrate the idea of a probabilistic module with variational inference. This idea can also be applied to other inference algorithms.

## 3.1 Probabilistic Module with Variational Inference

The concept of hiding the details of inference of a probabilistic module by specialized inference methods is nice. The challenge is that not all the approximate inference method are compatible with each other. In this section, we present an approach to implement probabilistic module with the framework of variational inference.

The main idea of variational inference is to approximate an intractable posterior distribution of latent variables with a parametric approximation, referred to as a variational posterior distribution. VI is often framed as a lower bound of the logarithm of the marginal distribution, e.g,

$$\log p(y|z) = \log \int_x p(y|x)p(x|z) \geq \int_x q(x|y,z) \log \frac{p(y|x)p(x|z)}{q(x|y,z)} = \mathcal{L}(y,z), \tag{1}$$

where $p(y|x)p(x|z)$ forms a probabilistic model with $x$ as a latent variable, $q(x|y)$ is the variational posterior distribution, and the lower bound is denoted as $\mathcal{L}(y,z)$. By then taking a natural exponentiation of $\mathcal{L}(y,z)$, we get a lower bound of the marginal probability denoted as $\tilde{p}(y|z) = e^{\mathcal{L}(y,z)}$.

Assume we are interested in plugging $p(y|z)$ into another probabilistic model $p(l|y)p(y|z)$ where $y$ is a latent variable. With variational inference, the lower bound of the overall model can be derived as

$$\log p(l|z) \geq \int_y q(y) \log \frac{p(l|y)p(y|z)}{q(y)} \geq \int_y q(y) \log \frac{p(l|y)\tilde{p}(y|z)}{q(y)}, \tag{2}$$

where the usual variational lower bound is further lower bounded by replacing $p(y|z)$ with its lower bound $\tilde{p}(y|z)$. By substituting (1) into (2), we then derive the variational lower bound for the whole model,

$$\log p(l|z) \geq \int_{y,x} q(x|y,z)q(y|z) \log \frac{p(l|y)p(y|x)p(x|z)}{q(x|y,z)q(y|z)} = \mathcal{L}(l,z). \tag{3}$$

This example shows that variational inference has a recursive property that enables *inference modularity*. A technical challenge with VI is that the integral of the lower bound of a probabilistic module

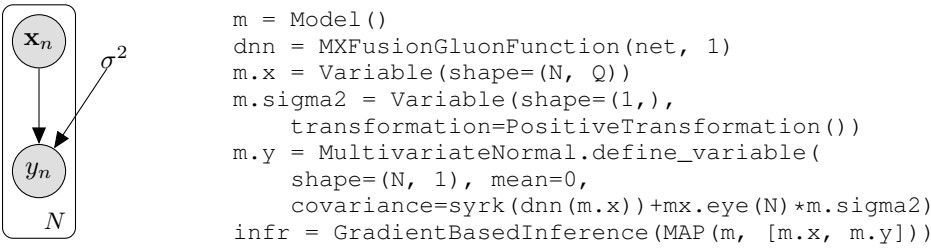

```
m = Model()
dnn = MXFusionGluonFunction(net, 1)
m.x = Variable(shape=(N, Q))
m.sigma2 = Variable(shape=(1,),
    transformation=PositiveTransformation())
m.y = MultivariateNormal.define_variable(
    shape=(N, 1), mean=0,
    covariance=syrk(dnn(m.x))+mx.eye(N)*m.sigma2)
infr = GradientBasedInference(MAP(m, [m.x, m.y]))
```

Figure 1: Bayesian linear regression. The variable `net` references a deep neural network defined in the MXNet Gluon syntax.

with respect to external latent variables, such as (2), may not always be tractable. Stochastic variational inference (SVI) offers an approximated solution to this new intractability by applying Monte Carlo Integration. Monte Carlo Integration is applicable to generic probabilistic distributions and lower bounds as long as we are able to draw samples from variational posterior.

In this case, the lower bound is approximated as

$$\mathcal{L}(l, z) \approx \frac{1}{N} \sum_i \log \frac{p(l|y_i)e^{\mathcal{L}(y_i, z)}}{q(y_i|z)}, \quad \mathcal{L}(y_i, z) \approx \frac{1}{M} \sum_j \log \frac{p(y_i|x_j)p(x_j|z)}{q(x_j|y_i, z)}, \quad (4)$$

where $y_i|z \sim q(y|z), x_j|y_i, z \sim q(x|y_i, z)$ and $N$ is the number of samples of $y$ and $M$ is the number of samples of $x$ given $y$. Note that if there is a closed form solution of $\tilde{p}(y_i|z)$, the calculation of $\mathcal{L}(y_i, z)$ can be replaced with the closed-form solution.

MXFusion offers modularity via probabilistic modules, which combine the definition of probabilistic distributions and specialized inference methods in a concise interface. If using VI as the primary inference algorithm, probabilistic modules used in a model automatically compute variational lower bounds, such as $\tilde{p}(y|z)$ in the above example.

## 4 MODEL CONSTRUCTION WITH PROBABILISTIC MODULES

In PPLs, a probabilistic model is often represented as a graph of random variables, where connections between random variables denotes corresponding probabilistic distributions. In such a representation, a probabilistic distribution is a fundamental re-usable component for constructing various probabilistic models. A probabilistic distribution typically have two reusable operations required by an inference method: drawing a sample of the random variable and computing the logarithm of its probabilistic density/mass function. Then, a user specifies a probabilistic model by defining individual variables and their corresponding probabilistic distributions if they are random variables. Each specified probabilistic distribution includes the form of the distribution and all the variables that the distribution is conditioned on, which corresponds to all the directed connections towards the random variable.

MXFusion follows this design of model specification. Figure 1 implements a Bayesian linear regression (BLR) model using a deep neural network as a feature extractor. There are $N$ data points, each of which contains an input vector $\mathbf{x}_n \in \mathbb{R}^Q$ and an output variable $y_n \in \mathbb{R}$. Each input vector is first applied to a DNN, $\mathbf{h}_n = f_\phi(\mathbf{x}_n)$, where $\phi$ denotes the parameters of the DNN. The output variable, $\mathbf{y} = (y_1, \ldots, y_N)$, follows a multi-variate normal distribution, $\mathbf{y} \sim \mathcal{N}(0, \mathbf{H}^\top \mathbf{H} + \sigma^2 \mathbf{I})$, where $\mathbf{H}$ is the stack of the outcome of DNN, $\mathbf{H} = (\mathbf{h}_1, \ldots, \mathbf{h}_N)^\top$. As there are no latent variables in BLR, a `MAP` inference method is created to estimate the model parameters with maximum likelihood given the observed data, i.e., $\phi^*, \sigma^* = \arg\max_{\phi,\sigma} \log p(\mathbf{y}|f_\phi(\mathbf{X}), \sigma)$. The maximum likelihood estimate is achieved through optimizing the log-likelihood via a gradient-based optimization algorithm. The log-likelihood of BLR is computed via the `log_pdf` method of the probabilistic distribution of the variable $\mathbf{y}$, which is a multi-variate normal distribution. The gradient of the log-likelihood is obtained through auto-differentiation using MXNet.

A noticeable difference from PyMC and Edward is that all the variables in a probabilistic model are defined as explicit members of a `Model` class, which offers a clean memory management by

```
m = Model()
dnn = MXFusionGluonFunction(net, 1)
m.x = Variable(shape=(N, Q))
m.h = dnn(m.x)
m.sigma2 = Variable(shape=(1,),
    transformation=PositiveTransformation())
m.y = SparseGaussianProcessRegression.define_variable(
    shape=(N, 1), X=m.h, kernel=RBF(100), noise_var=m.sigma2)
infr = GradientBasedInference(MAP(m, [m.x, m.y]))
```

Figure 2: Deep kernel learning (Wilson et al., 2016). The variable `net` references a deep neural network defined in the MXNet Gluon syntax, of the output dimensionality is 100.

avoiding any global states. Like Edward, the information about which variables are observed is not given until inference time. This allows reuse a model specification across multiple inference methods. When applying a machine learning model to a problem, one almost always uses a model with at least two inference methods that have different observed variables.: one for estimating model parameters/posterior distribution from training data and the other for making prediction with new data. Taking BLR as an example, during training, the input variable $\mathbf{X}$ and output variable $\mathbf{y}$ are both observed and the aim of inference is to estimate the model parameters that maximizes the log-likelihood, while at prediction stage, only the input variable $\mathbf{X}$ is observed and the inference is to estimate the output variable with the optimal model parameters $p(\mathbf{y}|\mathbf{X}, \sigma^*, \theta^*)$.

### 4.1 PROBABILISTIC MODULE AS MODEL COMPONENT

A stronger modularity for PPL is to enable reusing probabilistic models as building blocks. Ideally, one should be able to construct a probabilistic model by putting together well-known sophisticated probabilistic models such as Gaussian process, just like building a deep neural work with convolutional layers or LSTM layers. MXFusion offers such modularity by encapsulating the specification of a probabilistic model and a set of specialized inference algorithms, which is called a probabilistic module. A probabilistic module implements the same interface that a probabilistic distribution has, i.e., drawing a sample of output random variables and computing the logarithm of the probabilistic density/mass function. By following the same interface, a probabilistic module can be transparently used by a probabilistic model just like an usual probabilistic distribution.

In Figure 2, we demonstrate the convenience of a probabilistic module by modifying a BLR into deep kernel learning (DKL) (Wilson et al., 2016). This only requires to change one line of code, which replaces a multi-variate normal distribution with a sparse Gaussian process regression (SGPR) module. The syntax of specifying a probabilistic module is similar to the one for a normal distribution. The input arguments include the input variable to the SGPR module and the choice of kernel function for GP, which is radius basis function (RBF) in this case, and the noise variance for a Gaussian likelihood distribution. The function returns a variable representing the output of the SGPR module, which is assigned as the variable `m.y` in this example. In (Wilson et al., 2016), a particular GP approximation, known as KISS-GP, is used for scalability, which is replaced by a variational sparse GP approximation (Titsias, 2009) in this example.

Variational sparse GP speeds up the computation of log-likelihood by replacing it with a variational lower bound. It introduces a set of auxiliary variables $\mathbf{u} \in \mathbb{R}^M$, known as inducing data, and a set of corresponding inducing inputs $\mathbf{Z} \in \mathbb{R}^{M \times Q}$. The resulting variational lower bound is

$$\log p(\mathbf{y}|\mathbf{X}, \mathbf{Z}, \theta) \geq \mathcal{L}_{\text{SGP}}(\mathbf{y}, \mathbf{X}, \mathbf{Z}, \theta) = \int_{\mathbf{f},\mathbf{u}} p(\mathbf{f}|\mathbf{u}, \mathbf{X}, \mathbf{Z})q(\mathbf{u}) \log \frac{p(\mathbf{y}|\mathbf{f})p(\mathbf{f}|\mathbf{u}, \mathbf{X}, \mathbf{Z})p(\mathbf{u}|\mathbf{Z})}{p(\mathbf{f}|\mathbf{u}, \mathbf{X}, \mathbf{Z})q(\mathbf{u})}$$

where $q(\mathbf{f}, \mathbf{u}) = p(\mathbf{f}|\mathbf{u}, \mathbf{X}, \mathbf{Z})q(\mathbf{u})$ is the variational posterior. The above variational lower bound has a closed form solution with a computational complexity $O(NM^2)$, which is significantly lower than the cubic complexity of GP, $O(N^3)$. The `log_pdf` method of the SGPR module compute the above variational lower bound. The `MAP` inference method of the DKL model finds the parameters of SGPR $\theta$, $\mathbf{Z}$ and the parameters of DNN $\phi$ that maximize the lower bound of DKL, i.e., $\theta^*, \phi^*, \mathbf{Z}^* = \arg\max_{\theta,\phi,\mathbf{Z}} \mathcal{L}_{\text{SGP}}(\mathbf{y}, f_\phi(\mathbf{X}), \mathbf{Z}, \theta)$.

```
m = Model()
m.x = Normal.define_variable(mean=0, variance=1, shape=(N, Q))
m.sigma2 = Variable(shape=(1,), transformation=PositiveTransformation())
m.y = SparseGaussianProcessRegression.define_variable(
    shape=(N, D), X=m.x, kernel=RBF(Q), noise_var=m.sigma2)
q = Posterior(m)
q.mu = Variable(shape=(N, Q))
q.S = Variable(shape=(N, Q), transformation=PositiveTransformation())
q.x.assign_factor(Normal(mean=q.mu, variance=q.S))
infr = GradientBasedInference(SVI(m, q, [m.x, m.y]))
```

Figure 3: Bayesian Gaussian process latent variable model (Titsias & Lawrence, 2010).

The variational inference method of SGPR transparently provides a specialized implementation that exploits all the knowledge about the SGPR module and delivers good accuracy and efficiency. To achieve the same task with a standard PPL, one needs to explicit the prior distribution of a sparse Gaussian process regression model and the corresponding variational posterior. Even by doing so, the same quality of inference is not achievable with a generic variational inference method. Stochastic variational inference with reparameterization tricks would be the best choice in this case. The inference result will be slow and has high variance. This is because: (1) The inference algorithm is not able to figure out the cancellation of $p(\mathbf{f}|\mathbf{u}, \mathbf{X}, \mathbf{Z})$ in both the nominator and denominator. A direct evaluation of $p(\mathbf{f}|\mathbf{u}, \mathbf{X}, \mathbf{Z})$ results into a cubic complexity $O(N^3)$, which fails to deliver any speed-up comparing with GP. (2) The variational distribution $p(\mathbf{f}|\mathbf{u}, \mathbf{X}, \mathbf{Z})$ is high dimensional $\mathbb{R}^N$ and highly correlated. The variance of the Monte Carlo integration via sampling will be high. (3) The optimal value of $q(\mathbf{u})$ in the variational lower bound has a closed form solution, which is exploited by variational sparse GP (Titsias, 2009). With SVI, one needs to explicitly optimize it with gradient optimization. The speed and quality of the two inference approaches reflects the gap between a generic inference algorithm and a specialized inference algorithm. The probabilistic module aims at bridging this gap.

## 4.2 NESTED VARIATIONAL INFERENCE

A probabilistic module can be treated transparently as a probabilistic distribution. Then, it is straightforward to construct a probabilistic model consists of multiple probabilistic modules such as Deep GPs (Dai et al., 2016). In these models, some of the exposed variables of probabilistic modules is also latent variables. As shown in Section 3.1, as long as a variational inference method is used for the whole probabilistic model, the inference of individual probabilistic modules can be transparently handled.

Figure 3 implements Bayesian Gaussian process latent variable model (BGPLVM) (Titsias & Lawrence, 2010), which is an example of this kind of models. BGPLVM can be constructed by assign the input variable $\mathbf{X}$ a Gaussian distribution with zero-mean and unit-variance. As the input variable $\mathbf{X}$ is a latent variable, the marginal log-likelihood is not tractable anymore. A variational lower bound can be written as

$$\log p(\mathbf{Y}) \geq \int_{\mathbf{X}} q(\mathbf{X}) \log \frac{p(\mathbf{Y}|\mathbf{X})p(\mathbf{X})}{q(\mathbf{X})} \geq \int_{\mathbf{X}} q(\mathbf{X}) \log \frac{\mathcal{L}_{\text{SGP}}(\mathbf{y}, \mathbf{X}, \mathbf{Z}, \theta)p(\mathbf{X})}{q(\mathbf{X})}$$

where $p(\mathbf{Y}|\mathbf{X})$ is a GP and $q(\mathbf{X}) = \mathcal{N}(\mathbf{X}|\mu, \text{diag}(\mathbf{s}))$ is the variational posterior of $\mathbf{X}$, which is assumed to be a Gaussian distribution with a diagonal covariance matrix. By applying a variational sparse GP (Titsias, 2009) approximation to $p(\mathbf{Y}|\mathbf{X})$, we further lower bound the original lower bound by replace $p(\mathbf{Y}|\mathbf{X})$ with the variational sparse GP lower bound mentioned above $\mathcal{L}_{\text{SGP}}(\mathbf{y}, \mathbf{X}, \mathbf{Z}, \theta)$. For the expectation with respect to $q(\mathbf{X})$, we can apply stochastic variational inference (SVI) by drawing samples from $q(\mathbf{X})$. In this way, we result into a nested variational inference combining a generic inference method (SVI) with a specialized inference method (variational sparse GP). Following the same approach, it is also straight-forward to extend BGPLVM into a variationally auto-encoded GPLVM/deep GP (Dai et al., 2016) by parameterizing $\mu$ and $\mathbf{s}$ in $q(\mathbf{X})$ as the outcome of a DNN.

| Metric | Method | naval1 | naval2 | kin8nm | power |
|--------|--------|--------|--------|--------|-------|
| RMSE | SGP[†] (50) | 3.5e-5 (1.0e-5) | 3.1e-4 (9.0e-6) | 0.087 (3.1e-3) | 3.98 (0.19) |
| | SGP (50) | 3.7e-5 (1.1e-5) | 3.1e-4 (8.0e-6) | 0.089 (2.96e-3) | 3.98 (0.19) |
| | SGP (3200) | 0.6e-5 (0.6e-5) | 2.97e-4 (5.0e-6) | 0.068 (2.48e-3) | 3.08 (0.28) |
| | DKL (1000) | 2.0e-5 (1.2e-5) | 4.85e-4 (5.5e-4) | 0.062 (1.51e-3) | 3.39 (0.26) |
| TLL | SGP[†] (50) | 8.58 (0.22) | 6.66 (0.03) | 0.98 (0.02) | -2.80 (0.05) |
| | SGP (50) | 8.69 (0.19) | 6.67 (0.02) | 0.98 (0.02) | -2.80 (0.05) |
| | SGP (3200) | 10.70 (0.51) | 6.70 (0.02) | 1.28 (0.04) | -2.53 (0.10) |
| | DKL (1000) | 5.26 (4.35) | -83.8 (43.3) | -19.4 (1.73) | -12.7 (3.84) |

Table 1: Performance comparison of variational sparse GP (SGP) and deep kernel learning (DKL) on four standard regression benchmarks. SGP[†] is the GPy implementation. SGP is the SGP module in MXFusion. The implementation of DKL is shown in Figure 2. The numbers in the parentheses in the method column show the number of inducing points. Both rooted mean square error (RMSE) and test log-likelihood (TLL) are the mean of measure with 10-fold cross-validation. The number in the parentheses next to performance measure is its standard deviation.

## 5 EXPERIMENTS

In the previous examples, we demonstrate that various sophisticated probabilistic models can be easily constructed with a few lines of codes by using probabilistic modules. In this section, we evaluate the models that are constructed with probabilistic modules deliver good performance on real data.

### 5.1 GAUSSIAN PROCESS REGRESSION AND DEEP KERNEL LEARNING

We evaluate the variational sparse GP (SGP) (Titsias, 2009) module implemented in MXFusion and a variant of deep kernel learning (DKL) (Wilson et al., 2016) shown in Figure 2, comparing with a standard sparse GP implementation from GPy (2012). All the GP models use RBF kernels. By taking the advantage of GPU acceleration that is provided by MXNet, it allows to scale to a significantly higher number of inducing points. For DKL, we use one hidden layer with 20 hidden units and tanh activation. It outputs a 5-dimensional representation and, then, feeds into our SGP module.

In this experiment, we compare all the methods on four standard regression benchmarks. *Naval1* and *Naval2* are the same regression dataset with two different regression targets, they have 11934 data points and 16 dimensional inputs. *Kin8nm* has 8192 data points and 8 dimensional inputs. We used 90% data points for training and 10% for testing. All the datasets are normalized column-wise. We use 50 inducing points for GPy implementation which is a standard choice in the literature, and use 50 and 3200 inducing points for our SGP module. The Adam optimizer (Kingma & Ba, 2015) was used for training in MXFusion, while GPy models are trained with L-BFGS. Results with 10-fold averages measured in terms of root mean squared error (RMSE) and test set log likelihood (TLL) are provided in Table 1. GPy and MXFusion with 50 inducing points give similar performance for both RMSE and TLL. With 3200 inducing points, it gives significantly better performance on all the datasets for both RMSE and TLL. For DKL, with a relative small number of inducing points (1000), it produces a comparable RMSE with the help of a neural network, but does not give as good TLL.

## 6 DISCUSSION

Despite the expressiveness and flexibility provided by a PPL, a major limitation of PPL is that they rely on generic inference algorithms. Recent work such as Edward and Pyro support customizable inference algorithms, in particular, stochastic variational inference with a customized variational posterior. Once an inference algorithm is chosen, it remains the same across a probabilistic model. However, given a specific probabilistic model, e.g., a conjugate model, a specialized inference algorithm that exploits the mathematical properties of that particular model will always produce inference results that are as good or better than the generic inference in terms of both accuracy and efficiency. In practice, when doing probabilistic modeling there is a tradeoff between implementing a specialized inference algorithm for improved speed and performance at the cost of maintainability

and flexibility by introducing specialized code for each model. MXFusion aims at closing the gap between having specialized, highly performant algorithms and generic, easily maintained generic algorithms by introducing probabilistic modules. A probabilistic module consists of both a model and inference definition, defined together and wrapped up in a modular, plug-and-play package. This allows specialized inference algorithms for corresponding probabilistic modules to be smoothly integrated into the inference algorithm of a larger probabilistic model. By bringing modularity into probabilistic programming, MXFusion offers a flexible maintainable framework for doing probabilistic modeling while keeping the accuracy and efficiency of specialized inference algorithms.

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
