# OpenReview forum: "Modular Deep Probabilistic Programming"
_ICLR.cc/2019/Conference_

### Official Review · AnonReviewer2 · 2018-10-15
**A good work on modularisation of probabilistic programming languages**

**Rating:** 5
**Confidence:** 3

**Review:**

The paper presents an extension of the MXFusion language that allows the use of probabilistic modules. These modules are defined as a set of random variables and a specific probabilistic distribution. The modules also contain dedicated inference methods. Using these modules, one can use probabilistic distributions with inference methods tailored to the distribution, which are usually more efficient than generic inference systems.
The paper presents several examples using Gaussian process models, evaluated by comparison with GPy and the standard spare gaussian process method implemented in MXFusion.

Overall, the paper is well written and clear, and all claims are justified. The idea of modularization is not really new (as other systems implement something similar) but this approach tries to be general, in order not to pose constraints on the specification of modules. The related work section provides a good positioning of the approach.
I have not found any specific problems in the paper, the quality is rather high. However, the actual content of the paper describes an extension of an existing system. Such an extension is certainly important, but the paper does not provide much more information.
Moreover, the results of the experimental test do not seem to me to be able to support the main objective of the extension, which is to give the possibility to exploit more specific probabilistic model and inference methods to achieve better results than an approach using general methods.
As far as the execution of the system is concerned, is this extension able to improve the scalability or reduce the walltime? Is this visible in the presented test (at least in terms of speed up)? Or is the convenience of this approach the simpler way to define distributions?

As for minor issues that I can point out, one concerns the definition of shape in the Variable of m.sigma2 (figures 1, 2, 3). I do not know the used in MXFusion, thus this might not be an error, but it seems that in the shape definition something is missing. It is written that shape=(1,), is it correct or is there an error? In case of absence of error, what does the empty argument mean?

The power benchmark is not described.

In references, Thomas V. Wiecki is mentioned with and without the first letter of middle name. I suggest to uniform the references.

Typos:
- Abstract: "... but also sophisticated probabilistic model*s* such as ..."
- Sec. 1, first row of page 2: The sentence "this would bring the a lot of benefits ...". The "the" word should be deleted.
- Sec. 1 refers to a section after 4 which does not exist in the paper.
- Page 5: remove the full stop before the colon in the 4th row.
- Page 5: "The log_pdf method of the SGPR module compute*s* the above variational lower bound"
- Sec. 6: the sentence "MXFusion aims at closing the gap between having specialized, highly performant algorithms and generic, easily maintained generic algorithms by introducing probabilistic modules." should be corrected.


Pros
- The extension allows the use of modules that define specific probabilistic distribution/inference methods
- It seems easy to extend the system with other modules
- Its a really useful extension...

Cons
- The performance presented in the paper is not entirely convincing
- ... but it is just an extension of an existing system

---

> ### Author Response · Authors · 2018-11-26
> **Re: A good work on modularisation of probabilistic programming languages**
>
> Thanks for your suggestions and feedbacks.
>
> > the actual content of the paper describes an extension of an existing system. Such an extension is certainly important, but the paper does not provide much more information.
>
> We will include more details of the proposed approach.
>
> > the results of the experimental test do not seem to me to be able to support the main objective of the extension
>
> We will include more examples/experiments to show case our library.
>
> > As far as the execution of the system is concerned, is this extension able to improve the scalability or reduce the walltime? Is this visible in the presented test (at least in terms of speed up)? Or is the convenience of this approach the simpler way to define distributions?
>
> Our library provides a more convenient way to define sophisticated distributions. By taking the advantage of GPU acceleration, which is provided via the underlying MXNet library, the Gaussian process models implemented in our library is faster than the previous implementations in pure Python (in GPy).

---

### Official Review · AnonReviewer3 · 2018-11-02
**The main idea is the introduction of a new building block-probabilistic modules-into probabilistic programming with the aspiration to improve the modularity of the language.**

**Rating:** 4
**Confidence:** 4

**Review:**

The paper works with the modularization of PPLs with natural inspiration for the successful modularization recently introduced in all deep learning softwares.

* Within your so-called probabilistic modules you package dedicated inference methods that are tailored for this particular class of problems and argues that this will perform better than using a general purpose solver. For each specific case this does of course make a lot of sense. However, when it comes to the relevant case (especially within probabilistic programming) when we have a (often complex) combination of several probabilistic modules, how do you then leverage the tailored solvers? What is it that guarantees that these are relevant in the new combined construction?

* Related to the above you write in your conclusion that "Once an inference algorithm is chosen, it remains the same across a probabilistic model. However, given a specific probabilistic model, e.g., a conjugate model, a specialized inference algorithm that exploits the mathematical properties of that particular model will always produce inference results that are as good or better than the generic inference in terms of both accuracy and efficiency." This is of course true and it is also part of some existing PPLs, for example Birch via their so-called "delayed sampling":
http://proceedings.mlr.press/v84/murray18a/murray18a.pdf
The implementation there is very different from what you propose. As far as I can understand you require hard-coding of each specific model, whereas in the paper mentioned above they seem to automate att conjugate gradient calculations to a much greater extent. Why is it better to insist on hard-coding this for each probabilistic module? and how can you guarantee smooth functioning when several probabilistic modules are combined in complex ways?

* In the inference method that you briefly sketch in Section 3 you make use of VI and the intractable integrals that results are then handled using Monte Carlo. What is the gain of using VI + Monte Carlo compared to direct use of Monte Carlo? Via direct use of some kind of Monte Carlo method you would be able to guarantee performance and do proper analysis, whereas with VI you loose that capability. However, VI does of course have other pros, but my question arises due to the fact that you end up using Monte Carlo anyway.

* You write that "In PPLs, a probabilistic model is often presented as a graph of random variables...". This is certainly true and the word "often" is very important in this sentence. At the same time, is not one of the key reasons for using PPLs compared to probabilistic graphical models that it offers a richer model class compared to probabilistic graphical models? While I perfectly respect you choice to specifying models in MXFusion using using probabilistic graphical models I do find this quite restrictive and it seems to miss some of the key possibilities with PPLs.

* In your BLR example (which is very instructive by the way) you compute the solution via MAP. This is also find rather puzzling since that removes another great feature of PPLs, namely to work with probability distributions throughout the entire inference stage. The user can then of course choose to extract whatever point estimate might be needed in the end. Why do you remove this possibility by insisting on a specific point estimate? or is this just a particular choice of this example and not a general design choice?


The paper contains a lot of issues related to the use of the English language and would benefit from proper proofreading.

---

> ### Author Response · Authors · 2018-11-26
> **Re: The main idea is the introduction of a new building block-probabilistic modules-into probabilistic programming with the aspiration to improve the modularity of the language.**
>
> Thanks for your suggestions and feedbacks.
>
> > when we have a (often complex) combination of several probabilistic modules, how do you then leverage the tailored solvers? What is it that guarantees that these are relevant in the new combined construction?
>
> For variational inference, the guarantee for a combination of several probabilistic modules comes from the fact that, replacing a part of a prior distribution in a variational lower bound with another variational lower bound results into a further lower bound. Therefore, we can easily put together multiple probabilistic modules for a combined model. A good example will be deep Gaussian processes, e.g. (Damianou&Lawrence 2013), which can be constructed by putting together multiple variational sparse Gaussian processes.
>
> > This is of course true and it is also part of some existing PPLs, for example Birch via their so-called "delayed sampling". Why is it better to insist on hard-coding this for each probabilistic module? and how can you guarantee smooth functioning when several probabilistic modules are combined in complex ways?
>
> Thanks for pointing out the relevant work. We will cite it. In the case of conjugate models, the hard-coding approach would not be able to automatically choose the best approach. Instead, it relies on users to make the right choice. Although the hard-coding approach is not as smart, it is a more generic approach which can be applied to the cases beyond conjugate models.
>
> > At the same time, is not one of the key reasons for using PPLs compared to probabilistic graphical models that it offers a richer model class compared to probabilistic graphical models?
>
> PPLs can describe models beyond probabilistic graphical models, however, a big portion of real world use cases of PPLs are about Bayesian inference on probabilistic graphical models, e.g., Bayesian statistics with Stan. Our library starts with this restrictive use case and may extend in future.
>
> > Why do you remove this possibility by insisting on a specific point estimate? or is this just a particular choice of this example and not a general design choice?
>
> Doing point estimates in the example is a particular (common) choice of the model in this example. We will include examples with Bayesian inference on hyper-parameters.

---

### Official Review · AnonReviewer1 · 2018-11-02
**The paper proposes a new probabilistic programming language, but has a lack of scientific novelty**

**Rating:** 3
**Confidence:** 3

**Review:**

In this paper authors present a new Probabilistic Programming Language (PPL) MXFusion. Similarly to the languages for the deep learning (TensorFlow, PyTorch, etc.), this language introduce probabilistic modules that are used as building blocks for complex probabilistic models. Introducing modularity to the probabilistic programming, raises the problem of inference for probabilistic models. Since, we cannot obtain the exact solution on practice we have to resort to approximate inference methods. The approximate inference methods can be either generic, thus, being suitable for many probabilistic models but resulting in poor approximation, or specific, thus, having good approximation quality, but only for specific probabilistic models. Authors propose to address this problem by encapsulating specific inference methods in corresponding probabilistic modules. Doing so, one can perform approximate inference for every module with the best suitable inference technique. Authors demonstrate interface of MXFusion for three well known probabilistic models: Bayesian linear regression, deep kernel learning, Bayesian Gaussian process latent variable model.

Approaching the problem of building complex probabilistic models by introducing modular PPL is an important direction of study. But, regarding this paper I have the following concerns.
- In my opinion, the structure of the paper can be greatly improved. From general words about modularity and approximate inference authors dive to the very specific cases of probabilistic models. Following such structure, authors don’t give a clear answer to the following questions. Why the paradigm of encapsulating inference methods in probabilistic modules is legitimate for constructing complex probabilistic models? What inference methods and probabilistic models can we use as building blocks? Do we need to be aware of specific inference methods that are encapsulated or we can use any blocks in any order as we do in deep learning frameworks?
- Novelty of that paper is the new design of PPL. That is an interesting and important question for the community, but maybe ICLR paper is not the best format to present such kind of novelty.
- From the specific examples in the paper, legitimacy of such modular structure is clear only for variational inference (that seems to be a common knowledge) and variational approximation of gaussian processes. But the application area of MXFusion remains unclear. Verbatim examples of code for the specific examples doesn’t make the difference between MXFusion and other PPLs clear, because it can be treated as encapsulation of the code into some classes, that can be implemented in other languages as well.
- Comparison with other frameworks can be improved. In experimental section authors provide comparison with GPy framework in terms of RMSE and log-likelihood for gaussian process with 50 inducing points. As I understood both frameworks use the same inference methods and achieve the same performance, so the experiment can be considered as sanity check for MXFusion. The paper could benefit from comparison between different inference methods and providing benchmarks for inference time.

Overall, the paper proposes a new PPL that is an important direction of study, but have several drawbacks and conference paper format is not the best way to present such kind of novelty.

Typos:
- Page 1, “despite the different of DNNs…” -> “despite the difference of DNNs…”?
- Page 2, missing reference of the section
- Page 2, section 3, “... sightly different form.” -> “... slightly different form”?

---

> ### Author Response · Authors · 2018-11-26
> **Re: The paper proposes a new probabilistic programming language, but has a lack of scientific novelty**
>
> Thanks for the suggestions and feedbacks.
>
> > Why the paradigm of encapsulating inference methods in probabilistic modules is legitimate for constructing complex probabilistic models?
>
> A dedicated inference method for a specific model typically outperforms a generic black-box inference method. With the emphasis on flexibility, PPLs mostly build on generic inference methods such as black-box variational inference, which leads to a performance gap between the model with a dedicated implementation and the model implemented in a PPL. Encapsulating inference methods is an approach that can bridge the performance gap.
>
>
> > What inference methods and probabilistic models can we use as building blocks?
>
> We focused on variational inference in our library, but other inference methods can also be implemented under a similar idea, e.g., Rainforth (2018) proposes nested probabilistic programming for MCMC methods. For variational inference, probabilistic models that can benefit from an efficient variational lower bound are good candidates for building probabilistic modules.
>
> > Do we need to be aware of specific inference methods that are encapsulated or we can use any blocks in any order as we do in deep learning frameworks?
>
> By nesting a encapsulated variational lower bound into the external lower bound, the external inference method does not need to be aware of a specific choice of encapsulated inference method. In practice, a user may need to be aware of specific tuning parameters.
>
> > But the application area of MXFusion remains unclear.
>
> We will include better examples to show cases the difference.

---

### Public Comment · (anonymous) · 2018-09-30
**Somewhat important direction of research, but not entirely novel and lacking discussion**

The paper proposes a probabilistic programming (PP) framework with modular building blocks for deep learning (DL) models. Although suggested as novel, this is not new in the PP community. Model modularity is a fundamental to essentially all PP systems, and modular inference has been presented in a number of works [1-4]. Moreover, several frameworks such as Edward [1] and Pyro [5] are built specifically for DL models.

The authors focus on variational inference tactics for PP, yet this approach to inference in PP is well-demonstrated in Edward and others. A lesser focus is on Gaussian process (GP) models, which is an important direction of research in PP. The presented analysis on real world problems is useful and interesting. Yet the GP parts of this paper are lacking:
- Details of how a GP fits within the probabilistic programming framework is missing
- Should be more discussion of results

A couple important points regarding the submission:
- Authors should cite recent works in PP. For example, [3, 4] propose PP languages designed for modular inference, and [4] presents a probabilistic DSL designed for GPs. See https://probprog.cc/ for most recent work in the field.
- By naming the framework "MXFusion" in the paper the submission is no longer anonymized (see https://github.com/amzn/MXFusion#contributing)

[1] Tran et al. Edward: A library for probabilistic modeling, inference, and criticism. 2016.
[2] Mansinghka et al. Venture: a higher-order probabilistic
programming platform with programmable inference. 2014.
[3] Ge et al. Turing: a language for flexible probabilistic inference. 2018.
[4] Lavin & Mansinghka. Probabilistic programming for data-efficient robotics. 2018.
[5] Noah Goodman. Uber AI Labs open sources Pyro, a deep probabilistic programming language. 2017.

---

> ### Author Response · Authors · 2018-11-26
> **Re: Somewhat important direction of research, but not entirely novel and lacking discussion**
>
> Thanks for your comments.
>
> > Details of how a GP fits within the probabilistic programming framework is missing. Should be more discussion of results.
>
> We will include more details about the model in the examples and the experiments.
>
> > Authors should cite recent works in PP.
>
> Thanks for listing the relevant works. We will cite them.

---

### Meta-Review · Area_Chair1 · 2018-12-10
**modularity is good, but the specifics aren't really justified in relation to similar PPLs**

**Confidence:** 5
**Recommendation:** Reject

**Metareview:**

This paper presents a probabilistic programming language where models are constructed out of building blocks which specify both the distribution and an inference procedure. As a demonstration, they show how a GP-LVM can be implemented.

The paper spends a lot of space arguing for the benefits of modularity. Modularity is of course hard to argue with, and the benefits are already understood in the PPL community. But, as the reviewers point out, various other PPLs have already adopted various strategies to enable modular definition of models, and (in cases like Venture) special-purpose higher-level inference algorithms. This paper contains little discussion of other PPLs and how the specific design decisions relate to theirs, so it's hard to judge whether this paper really covers new ground. Such discussion wasn't added to the revised paper, even though multiple reviewers asked for it. I can't recommend acceptance.